# Mortality and Quality of Life with Chronic Kidney Disease: A Five-Year Cohort Study with a Sample Initially Receiving Peritoneal Dialysis

**DOI:** 10.3390/healthcare10112144

**Published:** 2022-10-28

**Authors:** Miquel Sitjar-Suñer, Rosa Suñer-Soler, Carme Bertran-Noguer, Afra Masià-Plana, Natalia Romero-Marull, Glòria Reig-Garcia, Francesc Alòs, Josefina Patiño-Masó

**Affiliations:** 1Primary Health Centre, Institut Català de la Salut, 17800 Olot, Spain; 2Nursing Department, University of Girona, 17003 Girona, Spain; 3Health and Health Care Research Group, Department of Nursing, University of Girona, 17003 Girona, Spain; 4Hospital Universitari Dr. Josep Trueta, 17007 Girona, Spain; 5Primary Health Centre, Passeig de Sant Joan, Institut Català de la Salut, 08010 Barcelona, Spain; 6Quality of Life Research Institute, University of Girona, 17003 Girona, Spain

**Keywords:** chronic kidney disease, renal replacement therapy, morbidity, mortality, perceived health, health-related quality of life

## Abstract

The quality of life, morbidity and mortality of people receiving renal replacement therapy is affected both by the renal disease itself and its treatment. The therapy that best improves renal function and quality of life is transplantation. Objectives: To study the quality of life, morbidity and mortality of people receiving renal replacement therapy over a five-year period. Design: A longitudinal multicentre study of a cohort of people with chronic kidney disease. Methods: Patients from the Girona health area receiving peritoneal dialysis were studied, gathering data on sociodemographic and clinical variables through an ad hoc questionnaire, quality of life using the SF-36 questionnaire, and social support with the MOS scale. Results: Mortality was 47.2%. Physical functioning was the variable that worsened most in comparison with the first measurement (*p* = 0.035). Those receiving peritoneal dialysis (*p* = 0.068) and transplant recipients (*p* = 0.083) had a better general health perception. The social functioning of transplant recipients improved (*p* = 0.008). Conclusions: People with chronic kidney disease had a high level of mortality. The dimension of physical functioning worsens over the years. Haemodialysis is the therapy that most negatively effects general health perception. Kidney transplantation has a positive effect on the dimensions of energy/vitality, social functioning and general health perception.

## 1. Introduction

The progressive ageing of the population and the resulting increase in life expectancy evince the paradigm shift of the welfare state promoted by advances in healthcare and current lifestyles. However, this longevity, associated with successes in public health policies and the socioeconomic development of states, is also associated with the proliferation of chronic diseases [1,2]. Among these diseases, end-stage renal disease and its associated risk factors are increasing worldwide [3], leading to an increased need for dialysis and kidney transplantation. Diverse factors are involved in the prevention of and approach to chronic kidney disease that depend on, among others, healthcare policies, human and economic resources, and cultural considerations [4].

During the current pandemic, health services have focused their care activity on controlling the infectious disease and the follow-up and prevention of chronic disease. Additionally, mobility has been regulated and contacts between different levels of care have been restricted in order to optimise resources and limit the spread of the virus [5,6].

Peritoneal dialysis is an ambulatory treatment that is more compatible with a wide range of daily life activities than haemodialysis. Its use has made it possible to be less dependent on healthcare systems during periods of confinement in the recent health crisis, contributing to a reduction in the transmissibility of the virus through hospital contact [7,8].

Although most studies are performed in people receiving haemodialysis, there are an increasing number of studies of peritoneal dialysis. Hiramatsu et al. [9] reported that patients who used this technique perceive greater benefits in the quality of life. The treatment has been found to have a particularly beneficial impact in terms of the dimensions that refer to problems and symptoms provoked by the health condition, the perception of burden, and the ability to continue working [10,11,12]. It should also be noted that it preserves residual renal function, permits a less restrictive diet, and conserves vascular access with regard to haemodialysis [13,14,15]. Furthermore, the cost to the health service of people using peritoneal dialysis is significantly lower than haemodialysis, showing that making an investment in and achieving good results with the initial treatment is beneficial for national health services [16,17]. Different authors coincide in affirming that the success of this dialysis is largely determined by the adherence to the treatment in terms of diet, medicines and therapies [18,19].

Despite the important role of dialysis as a therapy for chronic kidney disease, it has been demonstrated that kidney transplantation is the treatment that most improves renal function, is most efficient both in terms of survival and cost to the health service, and improves quality of life despite requiring a complex surgical intervention, pharmacological treatment and intensive medical follow-up [20,21].

In recent years, a variety of factors have been described as influencing the treatment of chronic diseases. Identifying the different elements is of interest to health professionals in order to be able to intervene in lifestyles and achieve efficacy in the treatment, especially in those areas that the patient is responsible for. Many studies agree that quality of life and social support, among other factors, interact with good adherence to treatment [22,23] and modulate the impact of chronic disease, which is related to an individual’s state of health. All of this contributes to the promotion of healthy behaviours and the control of chronic diseases, and have an influence in balancing adaptation, the prognosis, and the restoration of health [24,25]. However, it has been reported that with years of renal replacement therapy, those modulating factors that are the responsibility of the patient and which predispose to maintaining a good balance are weakened, leading to the failure of the technique [26,27]. Although the disease itself causes a significant increase in morbidity and mortality from the early stages, which affects a high percentage of people with this disease [28,29,30], several recent studies have reported improved survival rates for people with chronic kidney disease [31].

The hypotheses of the present study are:Chronic kidney disease and renal replacement therapy over the years have a strong negative influence on morbidity and mortality rates.People receiving kidney transplants have a better perception of health-related quality of life than people receiving renal replacement therapy through peritoneal dialysis or haemodialysis.

From these hypotheses, the following objectives were set:To study the rates of morbidity and mortality associated with a cohort of people receiving peritoneal dialysis after five years of evolution of the disease.To study the evolution in terms of quality of life in a cohort followed for five years receiving renal replacement therapy through peritoneal dialysis, haemodialysis and/or kidney transplantation.

## 2. Materials and Methods

A longitudinal multicentre study of a cohort of people with chronic kidney disease receiving renal replacement therapy over a five-year period.

### 2.1. Participants and Variables

The initial sample was formed between June and October 2015 by all people with chronic renal failure aged 18 and over who used peritoneal dialysis as a replacement kidney treatment at the only two centres in Girona Health Region that provide peritoneal dialysis: the Dr. Josep Trueta University Hospital of Girona and the Figueres Hospital of The Fundació Salut Empordà. The second sample included all people in the follow-up cohort between January and October 2020 undergoing renal replacement therapy through peritoneal dialysis, haemodialysis (when peritoneal dialysis was not feasible) or kidney transplantation. The variables studied are sociodemographic and related to quality of life.

### 2.2. Data Gathering Procedure

A specifically designed data collection logbook was used that grouped together sociodemographic and clinical data as well as data related to quality of life gathered through validated questionnaires. To obtain the data, participants were previously informed about the follow-up study and that participation was voluntary either during one of the programmed visits or by telephone. After agreeing to participate, the day and time for gathering the data was set, trying as best as possible to coincide with scheduled hospital visits but visiting patients’ homes when this was not possible. Clinical data was obtained from the medical history of the patients through the electronic platforms of the two centres.

The logbooks used to record information were number coded to identify the patients and maintain anonymity. Only the principal investigator knew the relationship between the number of the questionnaire and the person to whom it corresponded.

### 2.3. Instruments

In this study to evaluate quality of life and its evolution, the validated Spanish version (1.4) of the SF-36 questionnaire was used [32]. This instrument consists of the following dimensions that explore the areas of physical functioning (10 items), physical role limitations (four items), emotional role limitations (three items), social functioning (two items), mental health (five items), bodily pain (two items), energy/vitality (four items), and general health perceptions (five items). (Cronbach’s alpha: Physical functioning α = 0.9; Physical role α = 0.91; Bodily pain α = 0.86; General health α = 0.78; Vitality α = 0.8; Social functioning a = 0.74; Emotional role α = 0.91; Mental health α = 0.81) [33]. The use of validated instruments is one of the tools used to avoid information biases while obtaining participant data.

### 2.4. Statistical Analysis

Quantitative variables are expressed with the mean and the standard deviation or the median and the interquartile range, and categorical variables are expressed with the absolute frequency and the percentage. The Chi-squared test and/or the Fisher test were used to study associations between categorical variables. Quantitative variables were compared using the Student’s *t*-test. Correlations between quality of life and social support dimensions were performed with Spearman’s rho test. We used Cox regression model analysis, crude and adjusted by age. The authors performed a verification of the assumptions of the Cox model, testing proportionality with Kaplan–Meier curves and Schoenfeld residuals to check the proportional hazards assumption. Significance was set at *p* < 0.05 with a confidence interval of 95%.

### 2.5. Ethical Issues

The study was approved by the ethical committee of the Dr. Josep Trueta University Hospital of Girona (Acceptance code: IRC-Dialysis data: 27 April 2015) and the Research Committee of the Figueres Hospital of the Fundació Salut Empordà before beginning the study. The research was conducted with full respect for the personal protection legislation set out in Spanish organic law 3/2018 and the Declaration of Helsinki of the World Medical Association concerning ethical principles for medical investigations in human beings.

## 3. Results

### 3.1. Clinical Characteristics and Mortality of the Participants

A cohort of 55 patients receiving renal replacement therapy through peritoneal dialysis, haemodialysis, and/or transplantation over a five-year period was studied longitudinally. Mortality was 47.2% (*n* = 26) (Figure 1).

In 30.7% of cases the cause of death was due to problems related to the peritoneal catheter, including intestinal perforations and/or peritonitis resulting in septic shock (*n* = 8); 19.4% (*n*: 5) had a cardiac origin; 15.4% (*n* = 4) of deaths were due to multi-organ complications after kidney transplantation; 11.5% (*n* = 3) were due to neoplasms of pulmonary, cerebral and pancreatic origins; 11.5% (*n* = 3) were as a result of sudden death at home without a specific identified cause; 7.6% (*n* = 2) were due to patients stopping treatment following a significant deterioration in their functional and cognitive health until death; and 3.9% (*n* = 1) were due to strokes. After five years of follow-up, 72.4% (*n* = 21) of the surviving participants at the time of data collection were transplant recipients.

Table 1 shows the comparison of sociodemographic characteristics and morbidity between survivors and deceased patients. It is noteworthy that the average age of the group of deceased patients was almost 10 years older (*p* = 0.005). Women survived more than men (*p* = 0.044). With regard to the associated chronic diseases, the proportion of people with diabetes was significantly higher in the group of patients who died (*p* = 0.001), followed by heart disease (*p* = 0.071).

In the Cox regression crude analysis in relation to mortality, diabetic patients had lower odds of survival (hazard ratio 3.40, 95% confidence interval 1.52–7.61, *p* < 0.003). However, in the Cox model adjusted by age only a tendency to statistical significance was observed (hazard ratio 0.46, confidence interval 0.197–1.119, *p* = 0.088). No significant differences were found for survival for patients with hypertension (hazard ratio 0.96, 95% confidence interval 0.27–3.03, *p* = 0.872), nor in the adjusted model (hazard ratio 0.48, 95% confidence interval 0.13–1.72, *p* = 0.486), with heart disease (hazard ratio 1.88, 95% confidence interval 0.83–4.29, *p* = 0.128) nor in the adjusted model (hazard ratio 1.92, 95% confidence interval 0.84–4.40, *p* = 0.122). Finally, no significant difference was found for survival for patients with hypercholesterolaemia (hazard ratio 1.58, 95% confidence interval 0.69–3.59, *p* = 0.270). However, a tendency to statistical significance was observed in the adjusted model by age (hazard ratio 0.43, 95% confidence interval 0.18–1.01, *p* = 0.053) (Figure 2A,B).

### 3.2. Quality of Life of the Study Survivors

With regard to the dimension of quality of life of people who survived, physical functioning was the only variable that worsened in comparison with the baseline measurement (*p* = 0.035). In the case of the dimensions of vitality (*p* = 0.047) and social functioning (*p* = 0.017), more favourable scores were obtained in comparison with baseline values (Table 2).

When comparing the dimensions of SF-36 by the type of treatment (Table 3), it was observed in the follow-up that transplant recipients performed worse in the physical functioning dimension (*p* = 0.063). In terms of overall health, participants who continued to receive peritoneal dialysis (*p* = 0.068) and transplant recipients (*p* = 0.083) perceived better health with a tendency to significance than those who received haemodialysis. In terms of vitality, people who underwent a transplant performed better than those who received haemodialysis or continued with peritoneal dialysis (*p* = 0.007). It was also seen that with the months of follow-up, social functioning improved significantly in people who did not receive any type of dialysis (*p* = 0.008) (Table 3).

## 4. Discussion

This study found that the quality of life and morbidity of people receiving renal replacement therapy is affected both by the renal disease itself and the dialysis received over the years. A five-year follow-up analysis has been undertaken of people receiving peritoneal dialysis in the Girona Health Region to find out their morbidity, mortality and quality of life, and to compare these outcomes with other renal therapies such as haemodialysis when peritoneal dialysis failed and kidney transplants when patients were waiting to be transplanted.

With regard to the survival of the 55 patients who began in the study, mortality was 47.2% (*n*: 26). This percentage of mortality is between 14% and 16% lower than figures reported in studies of patients receiving haemodialysis treatment [34,35]. In this respect, a meta-analysis of mortality by the type of dialysis showed a higher rate in the group receiving peritoneal dialysis than the haemodialysis group, and this tendency was clear both in diabetic patients and those who had been receiving dialysis for longer periods [36]. There are several main causes of mortality in the present study. Peritoneal catheter complications accounted for 30.7% (*n*: 8) of deaths. Of the few studies that evaluated this problem, most focus on specific cases [37,38,39,40]. The review by Li, Ng and Mcintyre revealed that different complications due to systemic peritoneal inflammation complications were factors that increased mortality [41]. Ye et al. found that patients with peritoneal inflammation caused by infections associated with the insertion and manipulation of the peritoneal catheter had a 90–95% greater risk of death [42]. Finally, Raticzak et al. described a single case of intestinal perforation as having particularly severe complications with worse prognosis in patients receiving this type of dialysis [43]. Mortality with a cardiac origin was 19.4% (*n*: 5) in our study, which was lower than the 61.1% found in the multicentre study by Feng et al. [44] and the 29% reported in the systematic revision by Rhee, Chou and Kalantar-Zadeh [45].

In the case of deaths after transplantation, there has been a reduction of 20% in cases both early and late post-transplant in recent years according to Ying et al. [46]. The complexity of the surgical intervention in these patients and the associated comorbidity [47] brings a high risk of death [46]. Furthermore, several authors have reported the risk of dying from a cardiovascular cause in the first few months after transplantation (0–3 months) as being between 30% and 40% higher than in the general population and between 20% and 30% higher from infections, and in later phases the risk of dying as a result of a tumour from 10 years after transplantation onwards is between 10% and 20% higher than in the general population [46,48,49]. Concerning higher mortality in diabetics, with a tendency to statistical significance in our study, several studies have described diabetes as a predictive factor of mortality in patients with renal insufficiency [50]. Of particular interest is the recent systematic review by Copur et al. in 2021, who also reported high percentages of mortality in diabetic patients with renal insufficiency, was the observation that serum glycated albumin predicts all-cause mortality risk in dialysis patients with diabetes mellitus [51]. In the case of heart disease, different studies conclude that it is not directly related to renal disease, as in the present study, but is rather a consequence of the evolution of the primary cause which, as well as having renal affectation, produces damage to the heart [52,53,54]. Concerning the presence of dyslipidaemia, the statin treatment of participants with this diagnosis is probably protective, acting as a cardiovascular protector; however, these results cannot be confirmed due to the small sample size. 

With regard to the evolution of the quality of life among the survivors, to the best of our knowledge there are no other follow-up studies with an initial sample of peritoneal dialysis patients that we can compare our results to. In the present study, physical function is the only dimension that is found to be more affected as years go by. In the cohort of haemodialysis patients studied by Ishiwatari et al., only a drop in physical functioning was particularly noticeable after some years [55]. In the present study, the dimensions of vitality, social functioning and general health gave scores that were higher than they had been five years earlier. These results can be explained by the type of treatment that was initially given. As is seen in the systematic review by Budhram et al., those who received home-based renal replacement therapy obtained better scores in general health, vitality and social function dimensions [56].

The increase of the scores in some dimensions of quality of life at the end of the follow-up analysis, can in part be explained by the high percentage of transplant recipients. This phenomenon has also been described in other studies where quality of life increases after a transplant [57,58]. Despite the significant benefits of transplantation that are reported, De Pasquale et al. have pointed out that people submitted to kidney transplantation also have high levels of anxiety and depression, which have negative repercussions on the quality of life [59].

When comparing quality of life by the type of therapy received, the lower results in the physical functioning dimension of transplant recipients could be related to the previous years of dialysis [60,61] as well as the complications associated with the surgical intervention [62]. In this line, Rocha et al., 2020 also obtained lower scores in the general health and physical wellbeing variables [63]. Despite this, Zhang, Guo and Ming and other studies found higher scores in all quality of life dimensions in transplant recipients than those receiving any type of dialysis [20]. The group of patients receiving peritoneal dialysis and transplant recipients obtained better scores in our research in the general health dimension than patients receiving haemodialysis. This is in line with the findings of the systematic review and meta-analysis by Chuasuwan et al., who concluded that people receiving peritoneal dialysis had more favourable outcomes in the dimensions of symptoms, cognitive function, social interaction, and vitality in comparison with patients on haemodialysis [64]. Concerning the significant better scores obtained in vitality and social functioning of transplant patients in our study, it should be noted that several publications describe kidney transplantation as having significantly better effects than other treatments within the different quality of life variables [65,66].

### Limitations

On the one hand, the size of the sample should be considered as a possible limitation despite including the follow-up of all people receiving renal replacement therapy through peritoneal dialysis in the Girona Health Region. It seems likely that a greater sample size would have allowed significance to be reached where results were inconclusive. On the other hand, the design is observational rather than experimental, and so it is difficult to establish causal relationships with this number of participants, despite following the patients for five years. However, despite this, and taking into consideration that cohort studies also allow causal relationships to be established, this research has identified the quality of life, morbidity and mortality of people receiving renal replacement therapy and compared the results with the type of treatment in order to better understand the complexity of people who live with chronic kidney disease.

Regarding the practical implications, the findings of this study show that nurses should place great emphasis on specific health education so that patients and family members can recognise the main signs and symptoms of infection. The planning of a process of health education and individualised training focused on the aspects mentioned could lead to better health outcomes.

## 5. Conclusions

This study has found that mortality was 47.2% after five years of follow-up of patients receiving peritoneal dialysis. With regard to quality of life, among the participants who have completed the follow-up, physical functioning deteriorated from the baseline measurement. On comparing SF-36 dimensions by the type of treatment, transplanted patients had worse results in the physical functioning dimension. With regard to the general health dimension, those who continued with peritoneal dialysis or had received transplants perceived better health than those receiving haemodialysis. It should also be highlighted that having received a transplant is seen to have a positive incidence on vitality, social functioning and health perception when comparison is made with the previous year.

## Figures and Tables

**Figure 1 healthcare-10-02144-f001:**
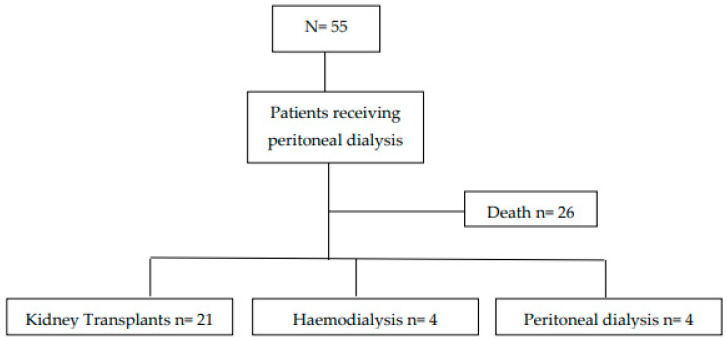
Evolution of patients and treatment used.

**Figure 2 healthcare-10-02144-f002:**
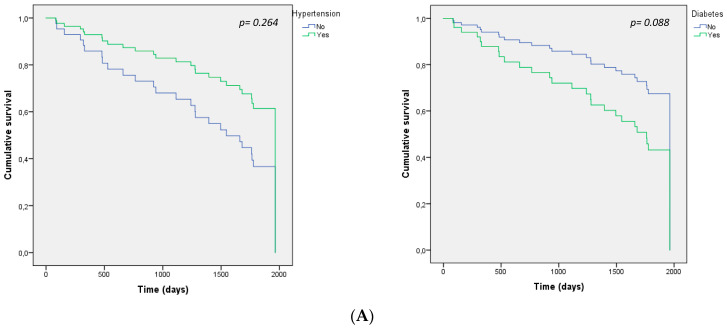
Cumulative survival of participants based on the presence of hypertension, diabetes, heart disease, or hyperlipidemia.

**Table 1 healthcare-10-02144-t001:** Comparison of the sociodemographic and clinical characteristics between survivors and deceased patients after five years of follow-up.

	Survivors *n*: 29	Deceased *n*: 26	*p*
**Age**	60.34 (11.601)	70.08 (11.185)	0.005
**Sex**			
Men	18 (32.7)	20 (36.4)	0.044
Women	11 (20)	6 (10.9)	
**Number of hospital admissions**	1.16 (1.675)	1 (1.063)	0.682
**Associated morbidity**			
Hypertension Yes	28 (50.9)	21 (38.1)	0.739
Diabetes Yes	6 (10.8)	15 (27.3)	0.001
Dyslipidaemia Yes	14 (25.4)	9 (16.4)	0.568
Heart disease Yes	5 (9)	9 (16.4)	0.071

Note: Quantitative variables are described with the mean and the standard deviation in parentheses, whereas qualitative variables are described with their absolute frequency and the corresponding percentage.

**Table 2 healthcare-10-02144-t002:** Comparison of SF-36 scores at baseline and follow-up in the group of survivors.

SF-36 Dimensions	Baseline Mean (SD) Median [IQR]	Follow-Up Mean (SD) Median [IQR]	*p*
**Physical functioning**	80.34 (15.11)	68.61 (27.05)	0.035
	85 [75–90]	80 [57.5–87.5]	
**Physical role**	56.03 (42.07)	67.24 (42.83)	0.431
	75 [0–100]	100 [12.5–100]	
**Bodily pain**	70.51 (26.07)	70.77 (27.29)	0.895
	77.5 [53.75–90]	70 [50–100]	
**General health**	42.93 (19.15)	47.24 (18.92)	0.391
	40 [30–55]	50 [35–60]	
**Vitality**	49.65 (21.08)	60.51 (21.84)	0.047
	50 [32.5–65]	60 [47.5–75]	
**Social functioning**	63.79 (28.41)	82.32 (30.16)	0.017
	62.5 [37.5–100]	100 [68.75–100]	
**Emotional role**	71.26 (41.52)	83.90 (35.20)	0.096
	100 [33.33–100]	100 [100–100]	
**Mental health**	67.58 (21.52)	74.75 (16.39)	0.152
	72 [46–84]	76 [66–86]	

Note: Wilcoxon test, non-parametric test.

**Table 3 healthcare-10-02144-t003:** Comparison of quality of life at baseline and follow-up by the type of renal replacement therapy received.

SF-36 Dimensions	Peritoneal Dialysis Mean (SD)	*p*	Haemodialysis Mean (SD)	*p*	Transplantation Mean (SD)	*p*
**Physical functioning**						
Baseline	81.25 (17.96)	0.705	61.25 (20.56)	0.273	83.80 (11.05)	0.063
Follow-up	78.75 (13.76)		38.75 (33)		72.38 (24.88)	
**Physical role**						
Baseline	62.50 (32.27)	0.109	62.50 (43.30)	0.157	53.57 (44.92)	0.334
Follow-up	100 (0)		25 (50)		69.04 (40.23)	
**Bodily pain**						
Baseline	83.75 (26.25)	0.713	58.12 (19.40)	1	70.35 (26.95)	0.904
Follow-up	80 (14.14)		52.50 (40.97)		72.50 (25.96)	
**General health**						
Baseline	67.50 (16.58)	0.068	45 (4.08)	0.713	37.85 (17.92)	0.083
Follow-up	55 (14.71)		43.75 (33.50)		46.42 (16.89)	
**Vitality**						
Baseline	72.50 (15.54)	0.109	45 (12.90)	0.854	46.19 (20.97)	0.007
Follow-up	58.75 (8.53)		43.75 (45.34)		64.04 (16.70)	
**Social functioning**						
Baseline	84.37 (18.75)	0.180	71.87 (21.34)	0.593	58.33 (29.66)	0.008
Follow-up	93.75 (7.21)		56.25 (51.53)		85.11 (26.40)	
**Emotional role**						
Baseline	91.66 (16.66)	0.317	83.33 (33.33)	0.317	65.07 (45.30)	0.101
Follow-up	100 (0)		75 (50)		82.53 (35.93)	
**Mental health**						
Baseline	78 (6.92)	0.785	73 (18.29)	0.414	64.57 (23.53)	0.204
Follow-up	79 (3.82)		81 (27.59)		72.76 (15.62)	

Note: Wilcoxon test, non-parametric test.

## Data Availability

The data presented in this study is available on request from the corresponding author. The data is not being made publicly available due to privacy considerations.

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
