# Peer review of "Mortality and Quality of Life with Chronic Kidney Disease: A Five-Year Cohort Study with a Sample Initially Receiving Peritoneal Dialysis"

_healthcare, 2022, doi:10.3390/healthcare10112144_

Round 1
Reviewer 1 Report
Thank you for your contributions toward understanding quality of life among chronic kidney disease patients.
This manuscript presents data from a longitudinal cohort study. The authors compare mortality rates among three kidney disease treatments. They also compare quality of life among the three treatments for the survivors of the study's cohort.
While the overall quality of the manuscript is good, the following recommendations are made for revision:
1. The title does not currently capture the study. The manuscript has a large emphasis on mortality, but mortality is not included in the title.
2. The results/discussion should be clarified. Perhaps I am misunderstanding, but it seems there are very few data comparisons with statistical significance? Yet the results narrative and discussion imply the comparisons were significant. This could be misleading or (at least) confusing to readers and should be clarified with precision.
3. The participants section explains that the initial sample was patients receiving peritoneal dialysis, and the cohort sample was patients receiving one of three treatments (peritoneal dialysis, hemodialysis, or transplant). I am assuming the difference between the samples is due to the peritoneal dialysis failing at some point in the 5 years? This should be clarified and also elaborated in the discussion.
4. The limitations section of the discussion states that the study was able to form causal relationships. This should be explained and clarified more precisely.
Author Response
Reviewer 1
Thank you for your contributions toward understanding quality of life among chronic kidney disease patients.
This manuscript presents data from a longitudinal cohort study. The authors compare mortality rates among three kidney disease treatments. They also compare quality of life among the three treatments for the survivors of the study's cohort.
We would like to thank you for your valuable comments.
While the overall quality of the manuscript is good, the following recommendations are made for revision:
- The title does not currently capture the study. The manuscript has a large emphasis on mortality, but mortality is not included in the title. Survival rate ¿? Mortality¿?
Reply: Thank you for your comment. We have incorporated the term "Mortality" in the title: “Mortality and quality of life with chronic kidney disease: a five-year cohort study with a sample initially receiving peritoneal dialysis”.
- The results/discussion should be clarified. Perhaps I am misunderstanding, but it seems there are very few data comparisons with statistical significance? Yet the results narrative and discussion imply the comparisons were significant. This could be misleading or (at least) confusing to readers and should be clarified with precision.
Reply: We have revised the results and discussion sections. To make the results section more understandable, we have added subtitles in this section: 3.1. Clinical Characteristics and mortality of the participants and 3.2. Quality of life of the study survivors. Significant data is shown with the p-value, for example when comparing the sociodemographic and clinical characteristics between survivors and deceased patients after five years of follow-up. Also, when comparing SF-36 scores at baseline and follow-up in the group of survivors or quality of life at baseline and follow-up by the type of renal replacement therapy received. We revised the discussion and rewrote some sentences as well as the limitations.
- The participants section explains that the initial sample was patients receiving peritoneal dialysis, and the cohort sample was patients receiving one of three treatments (peritoneal dialysis, haemodialysis, or transplant). I am assuming the difference between the samples is due to the peritoneal dialysis failing at some point in the 5 years? This should be clarified and also elaborated in the discussion.
Reply: It was exactly as you indicated. The cohort of patients with peritoneal dialysis at baseline were followed for five years. During the follow-up, a significant sample of patients died, as the results show, others received a kidney transplant because they were on the transplant waiting list, other participants continued with the same therapy or went on haemodialysis. Figure 1 shows the data on the evolution of this cohort.
- The limitations section of the discussion states that the study was able to form causal relationships. This should be explained and clarified more precisely.
Reply: We have rewritten the limitations section.
Reviewer 2 Report
I have read this manuscript carefully, and here are my comments:
Material and methods
The Cox model is a regression model for survival data. Although, the use of the Cox model without performing the verification of the assumptions is a common practice. The authors should detail if they committed a verification of the assumptions of the Cox model.
Results
The authors mention that diabetic patients had lower odds of survival (HR hazard ratio 3.40, 95% confidence interval 1.52-7.61, P<0.003), etc. Are these crude or adjusted HRs? Which were the factors of adjustment if this was the case? In the same paragraph, they refer to a Figure 2A and 2B. However, these figures are not shown in the manuscript.
References
All the references must be verified and rewritten according to the journal style.
Author Response
Reviewer 2
Material and methods
The Cox model is a regression model for survival data. Although, the use of the Cox model without performing the verification of the assumptions is a common practice. The authors should detail if they committed a verification of the assumptions of the Cox model.
Reply: The authors have performed a verification of the assumptions of the Cox model, testing proportionality with Kaplan-Meier curves and Schoenfeld residuals to check the proportional hazards assumption. This information has been written in the statistical analysis section of the methodology.
Results
The authors mention that diabetic patients had lower odds of survival (HR hazard ratio 3.40, 95% confidence interval 1.52-7.61, P<0.003), etc. Are these crude or adjusted HRs? Which were the factors of adjustment if this was the case? In the same paragraph, they refer to a Figure 2A and 2B. However, these figures are not shown in the manuscript.
Reply: The HR hazard ratio data was crude. Considering the comments, we have also realized the Cox regression model adjusted by age. In this sense, we have also written all the results of the model adjusted.
References
All the references must be verified and rewritten according to the journal style.
Reply: They have reviewed all the references and adjusted to the style that is required. The authors would like to thank you for your valuable comments.

Round 2
Reviewer 1 Report
I am pleased to see the revisions made by the authors and believe the data is presented clearly and accurately.
Reviewer 2 Report
I am satisfied with this latest version of the manuscript. Please check the references (they are double-numbered).